# TarMAC: Targeted Multi-Agent Communication

## Abstract

We explore a collaborative multi-agent reinforcement learning setting where a team of agents attempts to solve cooperative tasks in partially-observable environments. In this scenario, learning an effective communication protocol is key. We propose a communication architecture that allows for targeted communication, where agents learn both *what* messages to send and *who* to send them to, solely from downstream task-specific reward without any communication supervision. Additionally, we introduce a multi-stage communication approach where the agents co-ordinate via multiple rounds of communication before taking actions in the environment. We evaluate our approach on a diverse set of cooperative multi-agent tasks, of varying difficulties, with varying number of agents, in a variety of environments ranging from 2D grid layouts of shapes and simulated traffic junctions to complex 3D indoor environments. We demonstrate the benefits of targeted as well as multi-stage communication. Moreover, we show that the targeted communication strategies learned by agents are both interpretable and intuitive.

## 1 Introduction

Effective communication is a key ability for collaborative multi-agents systems. Indeed, intelligent agents (humans or artificial) in real-world scenarios can significantly benefit from exchanging information that enables them to coordinate, strategize, and utilize their combined sensory experiences to act in the physical world. The ability to communicate has wide-ranging applications for artificial agents – from multi-player gameplay in simulated games (*e.g.* DoTA, Quake, StarCraft) or physical worlds (*e.g.* robot soccer), to networks of self-driving cars communicating with each other to achieve safe and swift transport, to teams of robots on search-and-rescue missions deployed in hostile and fast-evolving environments.

A salient property of human communication is the ability to hold *targeted* interactions. Rather than the 'one-size-fits-all' approach of broadcasting messages to all participating agents, as has been previously explored (Sukhbaatar et al., 2016; Foerster et al., 2016), it can be useful to direct certain messages to specific recipients. This enables a more flexible collaboration strategy in complex environments. For example, within a team of search-and-rescue robots with a diverse set of roles and goals, a message for a fire-fighter ("smoke is coming from the kitchen") is largely meaningless for a bomb-defuser.

In this work we develop a collaborative multi-agent deep reinforcement learning approach that supports targeted communication. Crucially, each individual agent *actively selects* which other agents to send messages to. This targeted communication behavior is operationalized via a simple signature-based soft attention mechanism: along with the message, the sender broadcasts a key which encodes properties of agents the message is intended for, and is used by receivers to gauge the relevance of the message. This communication mechanism is learned implicitly, without any attention supervision, as a result of end-to-end training using a downstream task-specific team reward.

The inductive bias provided by soft attention in the communication architecture is sufficient to enable agents to 1) communicate agent-goal-specific messages (*e.g.* guide fire-fighter towards fire, bomb-defuser towards bomb, *etc.*), 2) be adaptive to variable team sizes (*e.g.* the size of the local neighborhood a self-driving car can communicate with changes as it moves), and 3) be interpretable through predicted attention probabilities that allow for inspection of *which* agent is communicating *what* message and to *whom*.

Our results however show that just using targeted communication is not enough. Complex real-world tasks might require *large populations of agents* to go through *multiple stages of collaborative communication and reasoning*, involving large amounts of information to be *persistent in memory* and exchanged via *high-bandwidth communication channels*. To this end, our actor-critic framework combines centralized training with decentralized execution (Lowe et al., 2017), thus enabling scaling to a large number of agents. In this context, our inter-agent communication architecture supports multiple stages of targeted interactions at every time-step, and the agents' recurrent policies support persistent relevant information in internal states.

While natural language, *i.e.* a finite set of discrete tokens with pre-specified human-conventionalized meanings, may seem like an intuitive protocol for inter-agent communication – one that enables human-interpretability of interactions – forcing machines to communicate among themselves in discrete tokens presents additional training challenges. Since our work focuses on machine-only multi-agent teams, we allow agents to communicate via continuous vectors (rather than discrete symbols), and via the learning process, agents have the flexibility to discover and optimize their communication protocol as per task requirements.

We provide extensive empirical demonstration of the efficacy of our approach across a range of tasks, environments, and team sizes. We begin by benchmarking multi-agent communication with and without attention on a cooperative navigation task derived from the SHAPES environment (Andreas et al., 2016). We show that agents learn intuitive attention behavior across a spectrum of task difficulties. Next, we evaluate the same targeted multi-agent communication architecture on the traffic junction environment (Sukhbaatar et al., 2016), and show that agents are able to adaptively focus on 'active' agents in the case of varying team sizes. Finally, we demonstrate effective multi-agent communication in 3D environments on a cooperative first-person point-goal navigation task in the rich House3D environment (Wu et al., 2018).

## 2   RELATED WORK

Multi-agent systems fall at the intersection of game theory, distributed systems, and Artificial Intelligence in general (Shoham & Leyton-Brown, 2008), and thus have a rich and diverse literature. Our work builds on and is related to prior work in deep multi-agent reinforcement learning, the centralized training and decentralized execution paradigm, and emergent communication protocols.

**Multi-Agent Reinforcement Learning (MARL).** Within MARL (see Busoniu et al. (2008) for a survey), our work is related to recent efforts on using recurrent neural networks to approximate agent policies (Hausknecht & Stone, 2015), algorithms stabilizing multi-agent training (Lowe et al., 2017; Foerster et al., 2018), and tasks in novel application domains such as coordination and navigation in 3D simulated environments (Peng et al., 2017; OpenAI, 2018; Jaderberg et al., 2018).

**Centralized Training & Decentralized Execution.** Both Sukhbaatar et al. (2016) and Hoshen (2017) adopt a fully centralized framework at both training and test time – a central controller processes local observations from all agents and outputs a probability distribution over joint actions. In this setting, any controller (*e.g.* a fully-connected network) can be viewed as implicitly encoding communication. Sukhbaatar et al. (2016) present an efficient architecture to learn a centralized controller invariant to agent permutations – by sharing weights and averaging as in Zaheer et al. (2017). Meanwhile Hoshen (2017) proposes to replace averaging by an attentional mechanism to allow targeted interactions between agents. While closely related to our communication architecture, his work only considers fully supervised one-next-step prediction tasks, while we tackle the full reinforcement learning problem with tasks requiring planning over long time horizons.

Moreover, a centralized controller quickly becomes intractable in real-world tasks with many agents and high-dimensional observation spaces (*e.g.* navigation in House3D (Wu et al., 2018)). To address these weaknesses, we adopt the framework of centralized learning but decentralized execution (following Foerster et al. (2016); Lowe et al. (2017)) and further relax it by allowing agents to communicate. While agents can use extra information during training, at test time, they pick actions solely based on local observations and communication messages received from other agents.

Finally, we note that fully decentralized execution at test time *without communication* is very restrictive. It means 1) each agent must act myopically based solely on its local observation and 2) agents cannot coordinate their actions. In our setting, communication between agents offers a rea-

|  | Decentralized Execution | Targeted Communication | Multi-Stage Decisions | Reinforcement Learning |
|---|---|---|---|---|
| DIAL (Foerster et al., 2016) | Yes | No | No | Yes (Q-Learning) |
| CommNets (Sukhbaatar et al., 2016) | No | No | Yes | Yes (REINFORCE) |
| VAIN (Hoshen, 2017) | No | Yes | Yes | No (Supervised) |
| ATOC (Jiang & Lu, 2018) | Yes | No | No | Yes (Actor-Critic) |
| TarMAC (this paper) | Yes | Yes | Yes | Yes (Actor-Critic) |

Table 1: Comparison with previous work on collaborative multi-agent communication with continuous vectors.

sonable trade-off between allowing agents to globally coordinate while retaining tractability (since the communicated messages are much lower-dimensional than the observation space).

**Emergent Communication Protocols.** Our work is also related to recent work on learning communication protocols in a completely end-to-end manner with reinforcement learning – from perceptual input (*e.g.* pixels) to communication symbols (discrete or continuous) to actions (*e.g.* navigating in an environment). While (Foerster et al., 2016; Jorge et al., 2016; Das et al., 2017; Kottur et al., 2017; Mordatch & Abbeel, 2017; Lazaridou et al., 2017) constrain agents to communicate with discrete symbols with the explicit goal to study emergence of language, our work operates in the paradigm of learning a continuous communication protocol in order to solve a downstream task (Sukhbaatar et al., 2016; Hoshen, 2017; Jiang & Lu, 2018). While (Jiang & Lu, 2018) also operate in a decentralized execution setting and use an attentional communication mechanism, their setup is significantly different from ours as they use attention to decide *when* to communicate, not *who* to communicate with ('who' depends on a hand-tuned neighborhood parameter in their work). Table 1 summarizes the main axes of comparison between our work and previous efforts in this exciting space.

## 3 TECHNICAL BACKGROUND

**Decentralized Partially Observable Markov Decision Processes (Dec-POMDPs).** A Dec-POMDP is a cooperative multi-agent extension of a partially observable Markov decision process (Oliehoek (2012)). For $N$ agents, it is defined by a set of states $S$ describing possible configurations of all agents, a global reward function $R$, a transition probability function $T$, and for each agent $i \in 1, ..., N$ a set of allowed actions $A_i$, a set of possible observations $\Omega_i$ and an observation function $O_i$. Operationally, at each time step every agent picks an action $a_i$ based on its local observation $\omega_i$ following its own stochastic policy $\pi_{\theta_i}(a_i|\omega_i)$. The system randomly transitions to the next state $s'$ given the current state and joint action $T(s'|s, a_1, ..., a_N)$. The agent team receives a global reward $r = R(s, a_1, ..., a_N)$ while each agent receives a local observation of the new state $O_i(\omega_i|s')$. Agents aim to maximize the total expected return $J = \sum_{t=0}^{T} \gamma^t r_t$ where $\gamma$ is a discount factor and $T$ is the episode time horizon.

**Actor-Critic Algorithms.** Policy gradient methods directly adjust the parameters $\theta$ of the policy in order to maximize the objective $J(\theta) = \mathbb{E}_{s \sim p_\pi, a \sim \pi_\theta(s)}[R(s, a)]$ by taking steps in the direction of $\nabla J(\theta)$. We can write the gradient with respect to the policy parameters as

$$\nabla_\theta J(\theta) = \mathbb{E}_{s \sim p_\pi, a \sim \pi_\theta(s)}[\nabla_\theta \log \pi_\theta(a|s) Q_\pi(s, a)],$$

where $Q_\pi(s, a)$ is called the action-value, it is the expected remaining discounted reward if we take action $a$ in state $s$ and follow policy $\pi$ thereafter. Actor-Critic algorithms learn an approximation of the unknown true action-value function $\hat{Q}(s, a)$ by e.g. temporal-difference learning (Sutton & Barto, 1998). This $\hat{Q}(s, a)$ is called the Critic while the policy $\pi_\theta$ is called the Actor.

**Multi-Agent Actor-Critic.** Lowe et al. (2017) propose a multi-agent Actor-Critic algorithm adapted to centralized learning and decentralized execution. Each agent learns its own individual policy $\pi_{\theta_i}(a_i|\omega_i)$ conditioned on local observation $\omega_i$, using a centralized Critic which estimates the joint action-value $\hat{Q}(s, a_1, ..., a_N)$.

## 4 TARMAC: TARGETED MULTI-AGENT COMMUNICATION

We now describe our multi-agent communication architecture in detail. Recall that we have $N$ agents with policies $\{\pi_1, ..., \pi_N\}$, respectively parameterized by $\{\theta_1, ..., \theta_N\}$, jointly performing a

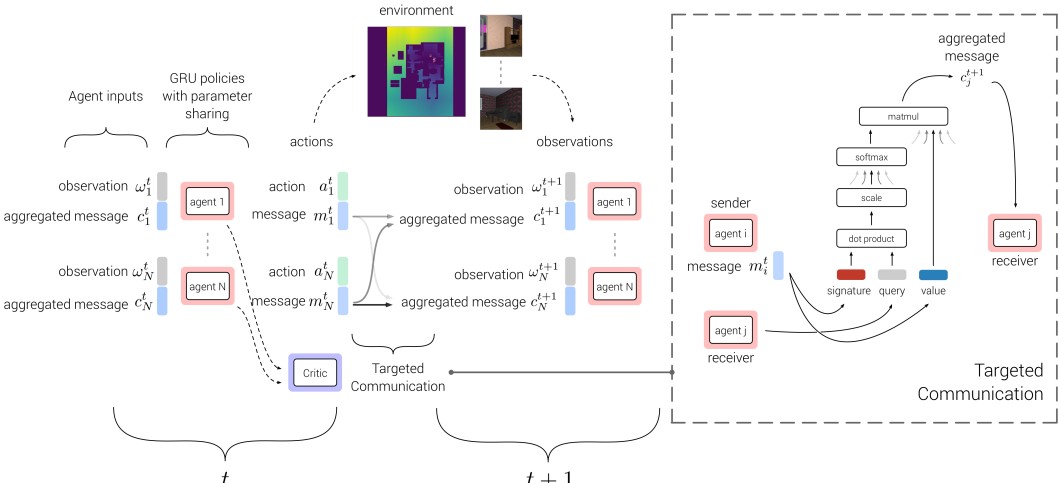

Figure 1: Overview of our multi-agent architecture with targeted communication. Left: At every timestep, each agent policy gets a local observation $\omega_i^t$ and aggregated message $c_i^t$ as input, and predicts an environment action $a_i^t$ and a targeted communication message $m_i^t$. Right: Targeted communication between agents is implemented as a signature-based soft attention mechanism. Each agent broadcasts a message $m_i^t$ consisting of a signature $k_i^t$, which can be used to encode agent-specific information and a value $v_i^t$, which contains the actual message. At the next timestep, each receiving agent gets as input a convex combination of message values, where the attention weights are obtained by a dot product between sender's signature $k_i^t$ and a query vector $q_j^{t+1}$ predicted from the receiver's hidden state.

cooperative task. At every timestep $t$, the $i$th agent for all $i \in \{1, ..., N\}$ sees a local observation $\omega_i^t$, and must select a discrete environment action $a_i^t \sim \pi_{\theta_i}$ and a continuous communication message $m_i^t$, received by other agents at the next timestep, in order to maximize global reward $r_t \sim R$. Since no agent has access to the underlying state of the environment $s_t$, there is incentive in communicating with each other and being mutually helpful to do better as a team.

**Policies and Decentralized Execution.** Each agent is essentially modeled as a Dec-POMDP augmented with communication. Each agent's policy $\pi_{\theta_i}$ is implemented as a 1-layer Gated Recurrent Unit (Cho et al., 2014). At every timestep, the local observation $\omega_i^t$ and a vector $c_i^t$ aggregating messages sent by all agents at the previous timestep (described in more detail below) are used to update the hidden state $h_i^t$ of the GRU, which encodes the entire message-action-observation history up to time $t$. From this internal state representation, the agent's policy $\pi_{\theta_i}(a_i^t \mid h_i^t)$ predicts a categorical distribution over the space of actions, and another output head produces an outgoing message vector $m_i^t$. Note that for all our experiments, agents are symmetric and policies are instantiated from the same set of shared parameters; *i.e.* $\theta_1 = ... = \theta_N$. This considerably speeds up learning.

**Centralized Critic.** Following prior work (Lowe et al., 2017; Foerster et al., 2018), we operate under the centralized learning and decentralized execution paradigm wherein during training, a centralized critic guides the optimization of individual agent policies. The centralized Critic takes as input predicted actions $\{a_1^t, ..., a_N^t\}$ and internal state representations $\{h_1^t, ..., h_N^t\}$ from all agents to estimate the joint action-value $\hat{Q}_t$ at every timestep. The centralized Critic is learned by temporal difference (Sutton & Barto, 1998) and the gradient of the expected return $J(\theta_i) = \mathbb{E}[R]$ with respect to policy parameters is approximated by:

$$\nabla_{\theta_i} J(\theta_i) = \mathbb{E}\left[\nabla_{\theta_i} \log \pi_{\theta_i}(a_i^t|h_i^t)\ \hat{Q}_t(h_1^t, ..., h_N^t, a_t^1, ..., a_t^N)\right].$$

Note that compared to an individual critic $\hat{Q}_i(h_i^t, a_i^t)$ for each agent, having a centralized critic leads to considerably lower variance in policy gradient estimates since it takes into account actions from all agents. At test time, the critic is not needed anymore and policy execution is fully decentralized.

**Targeted, Multi-Stage Communication.** Establishing complex collaboration strategies requires targeted communication *i.e.* the ability to send specific messages to specific agents, as well as multi-

stage communication *i.e.* multiple rounds of back-and-forth interactions between agents. We use a signature-based soft-attention mechanism in our communication structure to enable targeting. Each message $m_i^t$ consists of 2 parts – a signature $k_i^t \in \mathbb{R}^{d_k}$ to target recipients, and a value $v_i^t \in \mathbb{R}^{d_v}$:

$$m_i^t = [\; \overbrace{k_i^t}^{\text{signature}} \quad \underbrace{v_i^t}_{\text{value}} \;]. \tag{1}$$

At the receiving end, each agent (indexed by $j$) predicts a query vector $q_j^{t+1} \in \mathbb{R}^{d_k}$ from its hidden state $h_j^{t+1}$ and uses it to compute a dot product with signatures of all $N$ messages. This is scaled by $1/\sqrt{d_k}$ followed by a softmax to obtain attention weight $\alpha_{ji}$ for each message value vector:

$$\alpha_j = \text{softmax} \left[ \frac{q_j^{t+1^T} k_1^t}{\sqrt{d_k}} \; ... \; \underbrace{\frac{q_j^{t+1^T} k_i^t}{\sqrt{d_k}}}_{\alpha_{ji}} \; ... \; \frac{q_j^{t+1^T} k_N^t}{\sqrt{d_k}} \right] \tag{2}$$

$$c_j^{t+1} = \sum_{i=1}^{N} \alpha_{ji} v_i^t. \tag{3}$$

Note that equation 2 also includes $\alpha_{ii}$ corresponding to the ability to *self-attend* (Vaswani et al., 2017), which we empirically found to improve performance, especially in situations when an agent has found the goal in a coordinated navigation task and all it is required to do is stay at the goal, so others benefit from attending to this agent's message but return communication is not needed.

For multiple stages of communication, aggregated message vector $c_j^{t+1}$ and internal state $h_j^t$ are first used to predict the next internal state $h'_j^t$ taking into account a first round of communication:

$$h'_j^t = \tanh \left( W_{h \to h'} [\; c_j^{t+1} \parallel h_j^t \;] \right). \tag{4}$$

Next, $h'_j^t$ is used to predict signature, query, value followed by repeating Eqns 1-4 for multiple rounds until we get a final aggregated message vector $c_j^{t+1}$ to be used as input at the next timestep.

## 5 EXPERIMENTS

We evaluate our targeted multi-agent communication architecture on a variety of tasks and environments. All our models were trained with a batched synchronous version of the multi-agent Actor-Critic described above, using RMSProp with a learning rate of $7 \times 10^{-4}$ and $\alpha = 0.99$, batch size 16, discount factor $\gamma = 0.99$ and entropy regularization coefficient 0.01 for agent policies. All our agent policies are instantiated from the same set of shared parameters; *i.e.* $\theta_1 = ... = \theta_N$. Each agent's GRU hidden state is 128-d, message signature/query is 16-d, and message value is 32-d (unless specified otherwise). All results are averaged over 5 independent runs with different seeds.

### 5.1 SHAPES

The SHAPES dataset was introduced by Andreas et al. (2016)[1], and originally created for testing compositional visual reasoning for the task of visual question answering. It consists of synthetic images of 2D colored shapes arranged in a grid ($3 \times 3$ cells in the original dataset) along with corresponding question-answer pairs. There are 3 shapes (circle, square, triangle), 3 colors (red, green, blue), and 2 sizes (small, big) in total (see Figure 2).

We convert each image from SHAPES into an active environment where agents can now be spawned at different regions of the image, observe a $5 \times 5$ local patch around them and their coordinates, and take actions to move around – {up, down, left, right, stay}. Each agent is tasked with navigating to a specified goal state in the environment – {'red', 'blue square', 'small green circle', *etc.* } – and the reward for each agent at every timestep is based on team performance *i.e.* $r_t = \frac{\text{\# agents on goal}}{\text{\# agents}}$.

---

[1] github.com/jacobandreas/nmn2/tree/shapes

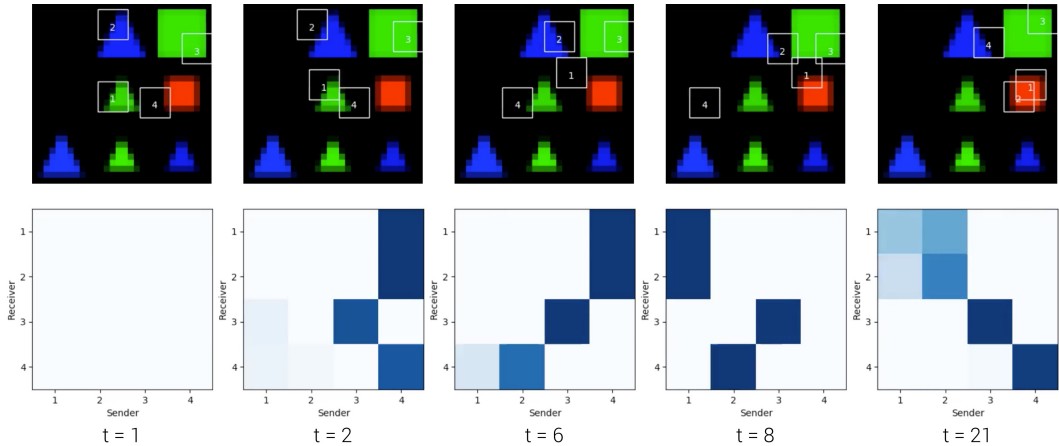

(a) 4 agents have to find [red, red, green, blue] respectively. $t = 1$: inital spawn locations; $t = 2$: 4 was on red at $t = 1$ so 1 and 2 attend to messages from 4 since they have to find red. 3 has found its goal (green) and is self-attending; $t = 6$: 4 attends to messages from 2 as 2 is on 4's target – blue; $t = 8$: 1 finds red, so 1 and 2 shift attention to 1; $t = 21$: all agents are at their respective goal locations and primarily self-attending.

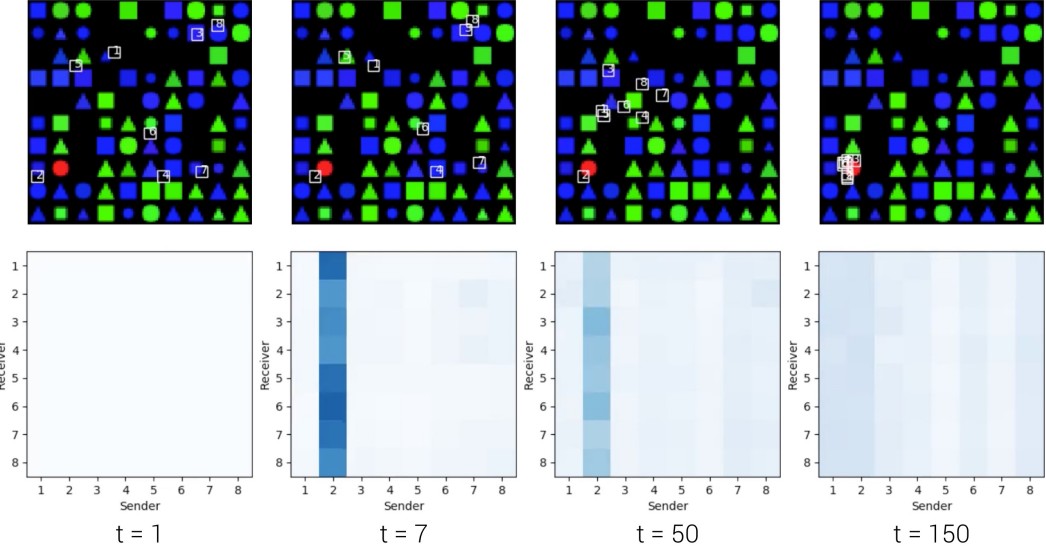

(b) 8 agents have to find red on a large $100 \times 100$ environment. $t = 7$: Agent 2 finds red and signals all other agents; $t = 7$ to $t = 150$: All agents make their way to 2's location and eventually converge around red.

Figure 2: Visualizations of learned targeted communication in SHAPES. Figure best viewed in color.

| | $30 \times 30$, 4 agents, find[red] | $50 \times 50$, 4 agents, find[red] | $50 \times 50$, 4 agents, find[red,red,green,blue] |
|---|---|---|---|
| No communication | 95.3±2.8% | 83.6±3.3% | 69.1±4.6% |
| No attention | **99.7**±0.8% | **89.5**±1.4% | 82.4±2.1% |
| TarMAC | **99.8**±0.9% | **89.5**±1.7% | **85.8**±2.5% |

Table 2: Success rates on 3 different settings of cooperative navigation in the SHAPES environment.

Having a symmetric, team-based reward incentivizes agents to cooperate with each other in finding each agent's goal. For example, as shown in Figure 2a, if agent 2's goal is to find red and agent 4's goal is to find blue, it is in agent 4's interest to let agent 2 know if it passes by red ($t = 2$) during its exploration / quest for blue and vice versa ($t = 6$). SHAPES serves as a flexible testbed for carefully controlling and analyzing the effect of changing the size of the environment, no. of agents, goal configurations, *etc*. Figure 2 visualizes learned protocols from two different configurations, and Table 2 reports quantitative evaluation for three different configurations. Benefits of communication and attention increase with task complexity ($30 \times 30 \rightarrow 50 \times 50$ & find[red] $\rightarrow$ find[red,red,green,blue]).

**How does targeting work in the communication learnt by TarMAC?** Recall that each agent predicts a signature and value vector as part of the message it sends, and a query vector to attend to incoming messages. The communication is targeted because the attention probabilities are a function of both the sender's signature and receiver's query vectors. So it is not just the receiver deciding how much of each message to listen to. The sender also sends out signatures that affects how much of each message is sent to each receiver. The sender's signature could encode parts of its observation most relevant to other agents' goals (for example, it would be futile to convey coordinates in the signature), and the message value could contain the agent's own location. For example, in Figure 2a, at $t = 6$, we see that when agent 2 passes by blue, agent 4 starts attending to agent 2. Here, agent 2's signature encodes the color it observes (which is blue), and agent 4's query encodes its goal (which is also blue) leading to high attention probability. Agent 2's message value encodes coordinates agent 4 has to navigate to, as can be seen at $t = 21$ when agent 4 reaches there.

## 5.2 TRAFFIC JUNCTION

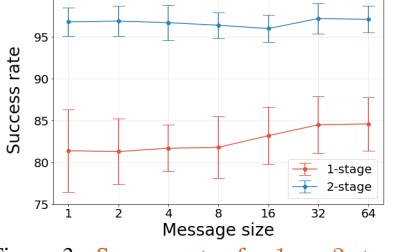

Figure 3: Success rates for 1 *vs.* 2-stage *vs.* message size on Hard. Performance does not decrease significantly even when the message vector is a single scalar, and 2 rounds of back-and-forth communication before taking an environment action leads to a significant improvement over 1-stage.

|  | Easy | Hard |
|---|---|---|
| No communication | $84.9_{\pm 4.3\%}$ | $74.1_{\pm 3.9\%}$ |
| CommNets (Sukhbaatar et al., 2016) | $99.7_{\pm 0.1\%}$ | $78.9_{\pm 3.4\%}$ |
| TarMAC 1-stage | $\mathbf{99.9}_{\pm 0.1\%}$ | $84.6_{\pm 3.2\%}$ |
| TarMAC 2-stage | $\mathbf{99.9}_{\pm 0.1\%}$ | $\mathbf{97.1}_{\pm 1.6\%}$ |

Table 3: Success rates on traffic junction. Our targeted 2-stage communication architecture gets a success rate of 97.1% on the 'hard' variant of the task, significantly outperforming Sukhbaatar et al. (2016). Note that 1- and 2-stage refer to the number of rounds of communication between actions (Equation 4).

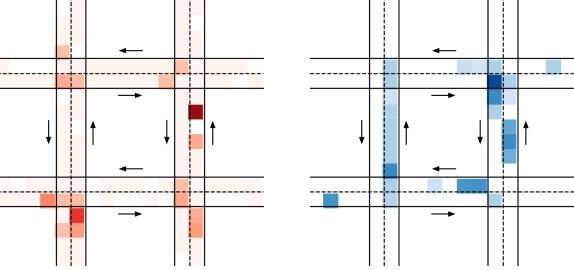

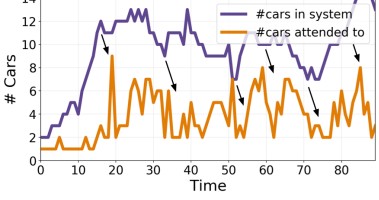

(a) Brake probabilities at different locations on the hard traffic junction environment. Cars tend to brake close to or right before entering junctions.

(b) Attention probabilities at different locations. Cars are most attended to in the 'internal grid' – right after the 1st junction and before the 2nd.

(c) No. of cars being attended to. 1) is positively correlated with total cars, indicating that TarMAC is adaptive to dynamic team sizes, and 2) is slightly right-shifted, since it takes few steps of communication to adapt.

Figure 4: Results on the traffic junction environment.

**Environment and Task.** The simulated traffic junction environments from Sukhbaatar et al. (2016) consist of cars moving along pre-assigned, potentially intersecting routes on one or more road junctions. The total number of cars is fixed at $N_{\max}$ and at every timestep, new cars get added to the environment with probability $p_{\text{arrive}}$. Once a car completes its route, it becomes available to be sampled and added back to the environment with a different route assignment. Each car has a limited visibility of a $3 \times 3$ region around it, but is free to communicate with all other cars. The action space for each car at every timestep is gas and brake, and the reward consists of a linear time penalty $-0.01\tau$, where $\tau$ is the number of timesteps since car has been active, and a collision penalty $r_{\text{collision}} = -10$.

**Quantitative Results.** We compare our approach with CommNets (Sukhbaatar et al., 2016) on the easy and hard difficulties of the traffic junction environment. The easy task has one junction of two one-way roads on a $7 \times 7$ grid with $N_{\max} = 5$ and $p_{\text{arrive}} = 0.30$, while the hard task has four connected junctions of two-way roads on a $18 \times 18$ grid with $N_{\max} = 20$ and $p_{\text{arrive}} = 0.05$.

See Figure 4a, 4b for an example of the four two-way junctions in the `hard` task. As shown in Table 3, a no communication baseline has success rates of $84.9\%$ and $74.1\%$ on `easy` and `hard` respectively. On `easy`, both CommNets and TarMAC get close to $100\%$. On `hard`, TarMAC with 1-stage communication significantly outperforms CommNets with a success rate of $84.6\%$, while 2-stage further improves on this at $97.1\%$, which is an $\sim18\%$ absolute improvement over CommNets.

**Model Interpretation.** Interpreting the learned policies, Figure 4a shows braking probabilities at different locations: cars tend to brake close to or right before entering traffic junctions, which is reasonable since junctions have the highest chances for collisions.

Turning our attention to attention probabilities (Figure 4b), we can see that cars are most-attended to when in the 'internal grid' – right after crossing the 1st junction and before hitting the 2nd junction. These attention probabilities are intuitive: cars learn to attentively attend to specific sensitive locations with the most relevant local observations to avoid collisions.

Finally, Figure 4c compares total number of cars in the environment *vs.* number of cars being attended to with probability $> 0.1$ at any time. Interestingly, these are (loosely) positively correlated, with Spearman's $\sigma = 0.49$, which shows that TarMAC is able to adapt to variable number of agents. Crucially, agents learn this dynamic targeting behavior purely from task rewards with no hand-coding! Note that the right shift between the two curves is expected, as it takes a few timesteps of communication for team size changes to propagate. At a relative time shift of 3, the Spearman's rank correlation between the two curves goes up to $0.53$.

**Message size *vs.* multi-stage communication.** We study performance of TarMAC with varying message value size and number of rounds of communication on the 'hard' variant of the traffic junction task. As can be seen in Figure 3, multiple rounds of communication leads to significantly higher performance than simply increasing message size, demonstrating the advantage of multi-stage communication. In fact, decreasing message size to a single scalar performs almost as well as 64-d, perhaps because even a single real number can be sufficiently partitioned to cover the space of meanings/messages that need to be conveyed for this task.

### 5.3 HOUSE3D

Finally, we benchmark TarMAC on a cooperative point-goal navigation task in House3D (Wu et al., 2018). House3D provides a rich and diverse set of publicly-available[2] 3D indoor environments, wherein agents do not have access to the top-down map and must navigate purely from first-person vision. Similar to SHAPES, the agents are tasked with finding a specified goal (such as 'fireplace'), spawned at random locations in the environment and allowed to communicate with each other and move around. Each agent gets a shaped reward based on progress towards the specified target. An episode is successful if all agents end within 0.5m of the target object in 50 navigation steps.

Table 4 shows success rates on a `find[fireplace]` task in House3D. A no-communication navigation policy trained with the same reward structure gets a success rate of $62.1\%$. Mean-pooled communication (no attention) performs slightly better with a success rate of $64.3\%$, and TarMAC achieves the best success rate at $68.9\%$. Figure 5 visualizes predicted navigation trajectories of 4 agents. Note that the communication vectors are significantly more compact (32-d) than the high-dimensional observation space, making our approach particularly attractive for scaling to large teams.

| | Success rate |
|---|---|
| No communication | $62.1_{\pm 5.3\%}$ |
| No attention | $64.3_{\pm 2.3\%}$ |
| TarMAC | $\mathbf{68.9}_{\pm 1.1\%}$ |

Table 4: Success rates on a 4-agent cooperative `find[fireplace]` navigation task in House3D.

## 6 CONCLUSIONS AND FUTURE WORK

We introduced TarMAC, an architecture for multi-agent reinforcement learning which allows targeted interactions between agents and multiple stages of collaborative reasoning at every timestep.

---

[2] github.com/facebookresearch/house3d

Figure 5: Agents navigating to the `fireplace` in House3D (marked in yellow). Note in particular that agent 4 is spawned facing away from it. It communicates with others, turns to face the `fireplace`, and moves towards it.

Evaluation on three diverse environments show that our model is able to learn intuitive attention behavior and improves performance, with downstream task-specific team reward as sole supervision.

While multi-agent navigation experiments in House3D show promising performance, we aim to exhaustively benchmark TarMAC on more challenging 3D navigation tasks because we believe this is where decentralized targeted communication can have the most impact – as it allows scaling to a large number of agents with large observation spaces. Given that the 3D navigation problem is hard in and of itself, it would be particularly interesting to investigate combinations with recent advances orthogonal to our approach (*e.g.* spatial memory, planning networks) with the TarMAC framework.

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
