# OpenReview forum: "TarMAC: Targeted Multi-Agent Communication"
_ICLR.cc/2019/Conference_

### Official Review · AnonReviewer1 · 2018-11-01
**Paper review**

**Rating:** 6
**Confidence:** 5

**Review:**

The authors present a study on multi-agent communication.
Specifically, they adapt communication to be targeted and multi-staged.
Experiments on  2 synthetic datasets and 1 3D visual dataset confirm that both additions are beneficial

Overall, this paper was somewhat clear and more importantly includes experiments on House3D, a more realistic dataset.

My main concern is the following: the method is not about targeting, but about selectively hearing.
If agents are sharing the reward then why should targeted communication be beneficial at all? Isn't the optimal strategy to just communicate everything to everyone? I understand that they should be selective at the listening side to properly integrate only the relevant information (so, attend over all received messages), but why should we expect the speaker to apriori know who this message should go to? Moreover, I don't really understand how targeted communication can even work (in the way the authors explain it) since the agents have partial information (e.g., in shapes they only see 5x5 around them), so they don't really know who is where --  but I could potentially see this working should the agents put information about their own identity and location.  So, given the positive results that the authors get, my understanding is that the signature doesn't have information about who should the recipient of the information be but more about what where the properties of the sender of this information.  So, based on my understanding, I don't feel that the flow of the story quite matches what is really happening and this might be very confusing for prospective readers. Can the authors elaborate on this, aim i getting things wrong?

There is literally no information about model size (or at least I wasn't able to find any). Is there any weight-sharing across agents? Do you obtain CommNets by using the implementations of the authors or by ablating the signature-part of your model? Moreover, why do agents have a limited view window on the SHAPES -- is (targeted) communication redundant when agents have full observability? The part about how multi-staged communication is implemented is quite cryptic at the moment -- is multi-staged the fact that the message is out-putted by processing with a recurrent unit? The messages is factorized into two parts k and u leading to a vector of size D -- what happens should we have one message of size D (rather than factorizing into 2), something like this would control for any improvements obtained from increases the parameters of the model.

Finally,  if the premises of the paper is to define more effective communication protocols, evident in the use of continuous communication, (rather than studying what form can multi-agent communication etc etc), a necessary baseline  (especially in cases where agents share reward), is to communicate the full observation (rather than a function of it).  This baseline is not presented here and it's absolutely necessary.

---

> ### Author Response · Authors · 2018-11-27
> **Responses to Reviewer 1 (part 1)**
>
> We thank the reviewer for their insightful feedback!
>
> > “My main concern is the following: the method is not about targeting, but about selectively hearing. If agents are sharing the reward then why should targeted communication be beneficial at all? Isn't the optimal strategy to just communicate everything to everyone? I understand that they should be selective at the listening side to properly integrate only the relevant information (so, attend over all received messages), but why should we expect the speaker to apriori know who this message should go to? Moreover, I don't really understand how targeted communication can even work (in the way the authors explain it) since the agents have partial information (e.g., in shapes they only see 5x5 around them), so they don't really know who is where -- but I could potentially see this working should the agents put information about their own identity and location. So, given the positive results that the authors get, my understanding is that the signature doesn't have information about who should the recipient of the information be but more about what where the properties of the sender of this information. So, based on my understanding, I don't feel that the flow of the story quite matches what is really happening and this might be very confusing for prospective readers. Can the authors elaborate on this, aim i getting things wrong?”
>
> In the SHAPES environment, in addition to a 5x5 image observation as input, the agents also get as input -- 1) an embedding of the goal they are supposed to navigate to, and 2) their own coordinates. 2 was missing from the description in the paper, we have added it in the revised version (apologies for this). So yes, agents do know where they are, and each agent is free to communicate any of this information with other agents.
>
> Each agent predicts three vectors for communication — signature, query, and value (Fig 1 and Eq 1). The communication is targeted because the attention probabilities are a function of both the sender’s signature and receiver's query vectors. So it is not just the receiver deciding how much of each message to listen to. That is, it is not just targeted listening. The sender also sends out signatures that affects how much of each message is sent to each receiver.
>
> The sender's signature could encode parts of its observation most relevant to other agents' goals (for example, it would be futile to convey coordinates in the signature). And the message value could contain the agent's own location. For example, in Fig 2a, we see that when agent 2 passes by blue, agent 4 starts attending to agent 2. Here, agent 2's signature encodes the color it observes (which is blue), and agent 4's query encodes its goal (which is also blue) leading to high attention probability. And agent 2's message value likely encodes coordinates for agent 4 to navigate to. We have included a discussion on this in Section 5.1.
>
> > “There is literally no information about model size (or at least I wasn't able to find any). Is there any weight-sharing across agents? Do you obtain CommNets by using the implementations of the authors or by ablating the signature-part of your model? “
>
> Each agent's GRU hidden state is 128-d, message signature/query is 16-d, and message value is 32-d (unless specified otherwise). We have updated section 5 with model size details (in orange). And yes, as mentioned in section 4, all agents share the same set of parameters.
>
> Results for CommNets are from their paper (https://arxiv.org/abs/1605.07736), and we benchmark our models on the same environment configurations as their paper using code obtained from the authors.
>
> > “Moreover, why do agents have a limited view window on the SHAPES -- is (targeted) communication redundant when agents have full observability?”
>
> Yes, communication is not needed when agents have full observability. In SHAPES and House3D, agents would know where the goal is, and in Traffic Junction, they would know the position of every other car, so they can navigate and maximize reward without having to communicate.
>
> > “The part about how multi-staged communication is implemented is quite cryptic at the moment -- is multi-staged the fact that the message is outputted by processing with a recurrent unit?”
>
> Multi-stage communication refers to the fact that agents are allowed to aggregate and exchange messages multiple times before taking one action in the environment. Concretely, Eq 4 is used to compute an updated hidden state for each agent from the aggregated message at previous timestep, followed by repeating Eq 1-3 to perform the next round of exchange of messages.

---

> ### Author Response · Authors · 2018-11-27
> **Response to Reviewer 1 (part 2)**
>
> > “The messages is factorized into two parts k and u leading to a vector of size D -- what happens should we have one message of size D (rather than factorizing into 2), something like this would control for any improvements obtained from increases the parameters of the model.”
>
> See figure 3, where we compare effect of increasing message size (i.e. adding more parameters) vs. multiple rounds of communication. Although this is still with the factorization into two parts, it captures change in performance with increase in model parameters. We find that simply increasing message size has little change in performance, and most of the gains come from multiple rounds of communication.
>
> > “Finally, if the premises of the paper is to define more effective communication protocols, evident in the use of continuous communication, (rather than studying what form can multi-agent communication etc etc), a necessary baseline (especially in cases where agents share reward), is to communicate the full observation (rather than a function of it). This baseline is not presented here and it's absolutely necessary.”
>
> On all 3 tasks studied in this work, a setting where each agent communicates its complete observation as the message, performs as well as TarMAC, and both outperform no attention (i.e. mean pooling messages). This is expected, since our environments are perceptually less complex than real-world scenarios. In principle, learning to communicate a function of the observation as in TarMAC allows compact representations of the observation to be transmitted, which is desirable in high-dimensional real-world observation spaces, where it would be infeasible and/or expensive to communicate complete observations, for example, a network of cars perceiving through a host of sensors, or a team of robots playing soccer.

---

> ### Author Response · Authors · 2018-11-29
> **Request for feedback**
>
> Hi Reviewer1 — thank you once again for your feedback on our work! We were wondering if you had any updated thoughts / feedback / questions following our response. We'd be happy to address additional concerns (if any). Please let us know either way. Thanks!

---

### Official Review · AnonReviewer2 · 2018-11-01
**An interesting extension of the 'learning to communicate' work using targeted messages and multiple rounds of communication.**

**Rating:** 6
**Confidence:** 4

**Review:**

The authors propose a new architecture for learning communication protocols. In this architecture each message consists of a key and a value. When receiving the message the listener produces an attention key that is used to selectively attend to some messages more than other using soft attention. This differs from the typical 'broadcasting' protocols learned in literature.

Questions / Comments:
- Eqn (4) looks like a vanilla RNN. Did you experience any issues around exploding or vanishing gradients when doing multiple rounds of communication? Why not use a gated architecture here?
- "Centralized Critic" section: This equation is from the COMA paper, ie. a centralised critic with policy gradients rather than DDPG. What did you use for the variance reduction baseline to estimate the advantage? Also, did you try conditioning the critic on the central state rather than the concat of observations? Formally this is required for the algorithm to be convergent.
- How many independent seeds are the results averaged over?
- The attention mechanism seems to provide very little value across all experiments:
-- 84.9% vs 82.7%
-- 89.5% vs 89.6%
-- 64.3% vs 68.9%
Did you check if any of these numbers are significant? This is my single biggest concern with the paper. Currently it's unclear whether attention is required at all in the settings presented. It would be good to see eg. the TarMAC 2-stage on the traffic junction (97.1%) ablated without attention.

---

> ### Author Response · Authors · 2018-11-27
> **Response to Reviewer 2**
>
> We thank the reviewer for their insightful feedback!
>
> > “Eqn (4) looks like a vanilla RNN. Did you experience any issues around exploding or vanishing gradients when doing multiple rounds of communication? Why not use a gated architecture here?”
>
> Yes, for a fair comparison to CommNets (Sukhbaatar et al., 2016) we use a formulation similar to a vanilla RNN. Indeed, gated recurrent units and other techniques can be employed to stabilize training in RNNs in multi-stage communication and would be interesting to explore in the future. This is orthogonal to the goal of this work though -- which is to develop a simple inter-agent targeting mechanism through attention. Moreover, our vanilla network trains fairly reliably for the 3 tasks we studied in our work.
>
> > “"Centralized Critic" section: This equation is from the COMA paper, ie. a centralised critic with policy gradients rather than DDPG. What did you use for the variance reduction baseline to estimate the advantage? Also, did you try conditioning the critic on the central state rather than the concat of observations? Formally this is required for the algorithm to be convergent.”
>
> Thanks for the pointer, we’ve cited both in the revised version. The equation corresponds to equation 4 from Lowe et al., 2017 (https://arxiv.org/abs/1706.02275) as well. Following Lowe et al., 2017, we do not condition the critic on the global state, but only on joint observations of all agents, and we do not use a variance reduction baseline. We will experiment with conditioning the critic on global state in future.
>
> > “How many independent seeds are the results averaged over? Did you check if any of these numbers are significant? This is my single biggest concern with the paper. Currently it's unclear whether attention is required at all in the settings presented.”
>
> All results are averaged over 5 independent runs with different seeds. The revised version has standard errors for all results. And yes, all the discussed trends are significant, i.e. wherever our submission claimed superior performance over no-attention across all 3 tasks (Table 2-4), they still hold.

---

> ### Author Response · Authors · 2018-11-29
> **Request for feedback**
>
> Hi Reviewer2 — thank you once again for your feedback on our work! We were wondering if you had any updated thoughts / feedback / questions following our response. We'd be happy to address additional concerns (if any). Please let us know either way. Thanks!

---

### Official Review · AnonReviewer3 · 2018-11-02
**Interesting extensions for multi-agent communication, it misses some baselines to illustrate the benefits of the contribution.**

**Rating:** 6
**Confidence:** 5

**Review:**

The authors present a multi-agent communication architecture where, agents can use targeted communication and can perform multiple communication steps. The paper is well written and easy to follow.

Comments:

1) The idea of multi-stage communication is great, but the paper doesn't have a strong point to support this contribution. Could the authors illustrate the benefit of multi-stage e.g. vs. the communication channel width?

2) In DIAL, the authors introduce a "null" action, what is the difference of that and multi-stage?

3) It is not clear to the reader what is the contribution of targeted communication vs. non-targeted as it looks a solution to the mean-pooling. Could the authors include at least one more experiment with on an architecture that doesn't use mean pooling. From an architecture perspective there is a scalability benefit of using pooling, but if that's the only one it has to be made more clear.

4) Following (3) based on Reddit there was a recent code release in python https://github.com/minqi/learning-to-communicate-pytorch. An alternative would be to evaluate TarMAC to one of the test beds, but the paper misses baselines.

---

> ### Author Response · Authors · 2018-11-27
> **Response to Reviewer 3**
>
> We thank the reviewer for their insightful feedback!
>
> > “1) The idea of multi-stage communication is great, but the paper doesn't have a strong point to support this contribution. Could the authors illustrate the benefit of multi-stage e.g. vs. the communication channel width?”
>
> We evaluated TarMAC on the hard variant of the traffic junction task with 1-stage and 2-stage communication and varying message value sizes. As can be seen in figure 3 in the revised draft, multiple rounds of communication leads to significantly higher performance than simply increasing message size, demonstrating the advantage of multi-stage communication. In fact, decreasing message size to a single scalar performs almost as well as when the message is 64-d (note that signature and query sizes were fixed at 32-d while we changed message value size), perhaps because even a single real number can be sufficiently partitioned to cover the space of meanings/messages that need to be conveyed for this task. We have added this discussion at the end of section 5.2.
>
> > “2) In DIAL, the authors introduce a "null" action, what is the difference of that and multi-stage?”
>
> Our understanding is that the reviewer is referring to the "None" action in the switch riddle game in DIAL. If that's the case, then the main difference is that the "None" action is an environment action that has an impact on the environment itself - whereas during our multi-stage communication, no environment actions are taken, but rather the agents are deliberating internally, sending back-and-forth messages multiple times before taking an environment action.
>
> > “3) It is not clear to the reader what is the contribution of targeted communication vs. non-targeted as it looks a solution to the mean-pooling. Could the authors include at least one more experiment with on an architecture that doesn't use mean pooling. From an architecture perspective there is a scalability benefit of using pooling, but if that's the only one it has to be made more clear. 4) Following (3) based on Reddit there was a recent code release in python https://github.com/minqi/learning-to-communicate-pytorch. An alternative would be to evaluate TarMAC to one of the test beds, but the paper misses baselines.”
>
> The "No attention" baselines in tables 2 and 4, and CommNets in table 3 all rely on mean-pooling, as opposed to TarMAC, which makes use of attentional pooling. TarMAC outperforms all mean-pooling baselines across SHAPES, Traffic Junction, and House3D. Results for CommNets are from their paper (https://arxiv.org/abs/1605.07736), and we benchmark our models on the same environment configurations as their paper using code obtained from the authors.
>
> The learnt communication is targeted because the attention probabilities are a function of both the sender’s signature and receiver's query vectors. So it is not just the receiver deciding how much of each message to listen to. That is, it is not just targeted listening. The sender also sends out signatures that affects how much of each message is sent to each receiver. For example in SHAPES, the sender can direct a message to “those looking for red objects” by encoding this in the signature. We have included a detailed discussion on this at the end of section 5.1 in the revised version.
>
> An architecture with no message pooling mechanism (attentional, mean, etc.) and with message concatenation instead has several crucial limitations -- 1) number of parameters scale linearly with number of agents, 2) no support for variable number of agents at training/test time -- both severely limiting scalability. For instance, in the traffic junction environment, the number of active cars in the system keeps changing across timesteps (violet curve in Fig 4c), so this experiment just cannot be run in this environment.
>
> So yes, TarMAC provides scalability benefits owing to attentional pooling -- by supporting a compact model size while allowing variable team sizes -- but also imparts intermediate interpretability to the communication channel through predicted attention probabilities, and outperforms mean-pooling across experiments.

---

> ### Author Response · Authors · 2018-11-29
> **Request for feedback**
>
> Hi Reviewer3 — thank you once again for your feedback on our work! We were wondering if you had any updated thoughts / feedback / questions following our response. We'd be happy to address additional concerns (if any). Please let us know either way. Thanks!

---

### Public Comment · ~Akshat_Agarwal1 · 2018-11-16
**How is communication targeted when each agent is receiving messages sent by all other agents and then deciding how much importance to give to each agent's message?**

This paper claims that the agents choose who to send messages to, however from Figure 1 and Section 4 it appears that each agent outputs a message=<signature,value> which is sent to ALL agents, who then use dot-product attention to give more or less weight (or importance) to messages from certain agents. So, each agent receives a message from all other agents, and then uses attention to aggregate these messages (instead of just taking a mean as done in CommNet, Sukhbaatar et al. 2016). Can you please elaborate on how the communication is targeted?

Another point: in 'VAIN: Attentional Multi-agent Predictive Modeling' (Hoshen 2017, published in NIPS 2017), each agent uses a similar attention mechanism to aggregate messages from other agents (however it uses an exponential kernel function instead of dot-product). Apart from the particular form of attention, can you please elaborate on the difference between your work and VAIN?

---

> ### Author Response · Authors · 2018-11-27
> **Response to question about 1) targeted communication, 2) comparison to VAIN**
>
> Thanks for your comments!
>
> > “This paper claims that the agents choose who to send messages to, however from Figure 1 and Section 4 it appears that each agent outputs a message=<signature,value> which is sent to ALL agents, who then use dot-product attention to give more or less weight (or importance) to messages from certain agents. So, each agent receives a message from all other agents, and then uses attention to aggregate these messages (instead of just taking a mean as done in CommNet, Sukhbaatar et al. 2016). Can you please elaborate on how the communication is targeted?”
>
> The learnt communication is targeted because the attention probabilities are a function of both the sender’s signature and receiver's query vectors. So it is not just the receiver deciding how much of each message to listen to. That is, it is not just targeted listening. The sender also sends out signatures that affects how much of each message is sent to each receiver.
>
> For example in SHAPES, the sender can direct a message to “those looking for red objects” by encoding this in the signature. We have included a detailed discussion on this at the end of section 5.1 in the revised version.
>
> > “Another point: in 'VAIN: Attentional Multi-agent Predictive Modeling' (Hoshen 2017, published in NIPS 2017), each agent uses a similar attention mechanism to aggregate messages from other agents (however it uses an exponential kernel function instead of dot-product). Apart from the particular form of attention, can you please elaborate on the difference between your work and VAIN?”
>
> Yes, VAIN proposes to replace averaging by a similar attentional mechanism to allow targeted interactions between agents. While closely related to our communication architecture, their work only considers fully supervised one-next-step prediction tasks, while we tackle the full reinforcement learning problem with tasks requiring planning over time horizons. Our submission already includes a discussion on this in section 2.

---

> > ### Public Comment · ~Akshat_Agarwal1 · 2018-11-29
> > **thanks for your response**
> >
> > 1) To the best of my understanding, this still means that all agents are sending a message to each other, and the attention weights depend both on a "signature" produced by the sender, and a "query" produced by the receiver. In my opinion, targeted communication would generally mean that the sender chooses a subset of agents to send a message to at each time step, while you seem to mean that the sender has an impact on how much attention the receiver chooses to pay to its message.  I think that was the primary confusion behind my question.
> > 2) Agreed.
> >
> > Thanks for your reply!

---

> > > ### Author Response · Authors · 2018-11-29
> > > **Response to comment on targeted communication**
> > >
> > > Yes, we think targeted communication implies targeting in both directions. Just the receiver deciding who to listen to would be targeted listening. Just the sender deciding who to send messages to would be targeted speaking/broadcasting. What we have is targeted two-way communication.

---

### Meta-Review · Area_Chair1 · 2018-12-19

**Confidence:** 4
**Recommendation:** Reject

**Metareview:**

The reviewers raised a number of concerns including the lack of clarity of various parts of the paper, lack of explanation, incremental novelty, and insufficiently demonstrated significance of the proposed. The authors’ rebuttal addressed some of the reviewers’ concerns but not fully. Overall, I believe that the paper presents some interesting extensions for multi-agent communication but in its current form the paper lacks explanations, comparisons and discussions. Hence, I cannot recommend this paper for presentation at ICLR.